# Efficient Adaptation of Pre-trained Vision Transformer via Householder Transformation

**Wei Dong[1], Yuan Sun[1], Yiting Yang[1], Xing Zhang[1], Zhijun Lin[2], Qingsen Yan[2],**
**Haokui Zhang[2], Peng Wang[3]\*, Yang Yang[3], Hengtao Shen[34]**
[1]College of Information and Control Engineering, Xi'an University of Architecture and Technology
[2]School of Computer Science, Northwestern Polytechnical University
[3]School of Computer Science and Engineering, University of Electronic Science and Technology of China
[4]School of Computer Science and Technology, Tongji University

## Abstract

A common strategy for Parameter-Efficient Fine-Tuning (PEFT) of pre-trained Vision Transformers (ViTs) involves adapting the model to downstream tasks by learning a low-rank adaptation matrix. This matrix is decomposed into a product of down-projection and up-projection matrices, with the bottleneck dimensionality being crucial for reducing the number of learnable parameters, as exemplified by prevalent methods like LoRA and Adapter. However, these low-rank strategies typically employ a fixed bottleneck dimensionality, which limits their flexibility in handling layer-wise variations. To address this limitation, we propose a novel PEFT approach inspired by Singular Value Decomposition (SVD) for representing the adaptation matrix. SVD decomposes a matrix into the product of a left unitary matrix, a diagonal matrix of scaling values, and a right unitary matrix. We utilize Householder transformations to construct orthogonal matrices that efficiently mimic the unitary matrices, requiring only a vector. The diagonal values are learned in a layer-wise manner, allowing them to flexibly capture the unique properties of each layer. This approach enables the generation of adaptation matrices with varying ranks across different layers, providing greater flexibility in adapting pre-trained models. Experiments on standard downstream vision tasks demonstrate that our method achieves promising fine-tuning performance.

## 1 Introduction

Parameter-Efficient Fine-Tuning (PEFT) for pre-trained Vision Transformers (ViTs) aims to adapt these models to downstream tasks by learning a small set of parameters while keeping most or all of the original model parameters frozen. This approach is expected to reduce the cost of fine-tuning and potentially improve the model's generalization performance, particularly when the downstream task involves limited data.

A common strategy for adapting the parameters is to learn an adaptation matrix that modifies the original matrix through addition or multiplication. To reduce the parameter scale of the adaptation matrix, a low-rank strategy is typically employed. This involves decomposing the adaptation matrix into the product of a down-projection matrix and an up-projection matrix, where the bottleneck dimensionality determines the parameter scale. Many prevailing PEFT solutions [1–3] follow this approach. However, these methods usually set the bottleneck dimensionality empirically to balance adaptation performance and parameter size. The fixed dimensionality, however, lacks the flexibility to accommodate variations in layer-wise properties.

---

*Corresponding author. Email address: p.wang6@hotmail.com

38th Conference on Neural Information Processing Systems (NeurIPS 2024).

In this work, we propose a novel parameter-efficient adaptation method to fine-tune pre-trained ViTs. Our design of the adaptation matrix is inspired by Singular Value Decomposition (SVD), which decomposes a matrix into a product of a left unitary matrix, a diagonal matrix, and a right unitary matrix. In SVD, the unitary matrices consist of orthogonal vectors, and the diagonal matrix of singular values essentially determines the rank of the matrix. Inspired by SVD, we propose to use Householder transformations to replace the left and right unitary matrices. Householder transformations maintain orthogonality properties similar to unitary matrices but can be derived simply by a vector, making them parameter efficient. With left and right Householder matrices, we learn the diagonal matrix adaptively for each layer to accommodate layer-wise properties. This approach, termed the Householder Transformation-based Adaptor (HTA), enables us to derive the adaptation matrix in a parameter-efficient manner while theoretically allowing for varying ranks for the adaptation matrices, thus achieving a better balance between parameter efficiency and adaptation performance.

We conducted experiments on a set of downstream vision classification tasks. The results show that our method can be effectively applied to various ViT versions, achieving promising fine-tuning performance. In summary, the contributions of this work can be summarized as follows:

- We approach PEFT from a novel angle by viewing the adaptation matrix from the perspective of SVD, which inspires us to propose a Householder transformation-based adaptation strategy that is parameter-efficient.
- By learning scaling coefficients to compose Householder transformation matrices together into adaptation matrices, our method can theoretically allow varying adaptation matrix ranks to accommodate layer-wise variations.
- Experiments on two sets of downstream vision classification tasks reveal our method can achieve an appealing balance between adaptation performance and parameter efficiency.

## 2 Related Work

### 2.1 Pre-training and Transfer Learning

As an advanced learning strategy, extensive research [4–7] has demonstrated the wide applicability of transfer learning across various fields. Especially in cases where the target task has limited data, high labeling costs, or poor data quality [8–10], transfer learning significantly enhances model generalization and training efficiency. By pre-training on large-scale datasets and using the obtained parameters as initialization for downstream tasks, transfer learning can effectively transfer and apply the knowledge of pre-trained models. In this process, the performance and convergence speed of downstream tasks are highly correlated with the dataset used for pre-training the model. In the field of computer vision, pre-training on large-scale datasets such as ImageNet [11] has significantly improved the performance of tasks like image classification [12–15], object detection [16, 17], and semantic segmentation [18, 19]. Moreover, self-supervised pre-training [20, 21] leverages the advantage of not requiring large amounts of labeled data, expanding the data scale and enhancing feature extraction capabilities, thereby further improving the generalization ability and robustness of pre-trained models. However, due to the substantial computational resources required to fully fine-tune the parameters of pre-trained models in downstream tasks, current research has shifted towards exploring more efficient fine-tuning methods.

### 2.2 Parameter-Efficient Fine-Tuning (PEFT)

Compared to full fine-tuning, the PEFT methods [22–25, 2, 26] aim to reduce the high cost of fine-tuning by freezing the majority of parameters in the pre-trained model and introducing a small number of learnable parameters to adapt to specific downstream tasks.

With the development of large pre-trained models, various PEFT approaches have emerged. Adapter [22] inserts a bottleneck-structured adapter layer into the pre-trained model and refines the model by updating only the parameters within the adapter layer. Bias [23] focuses on the fine-tuning of models for specific downstream tasks by meticulously calibrating the bias terms. VPT [24] integrates the concept of prompt learning into visual tasks, thereby facilitating targeted optimization for specific downstream tasks. SSF [25] efficiently fine-tunes the weights in pre-trained models

through affine transformations composed of scaling and shifting operations. AdaptFormer [2] explores a parallel adapter solution on ViT for various downstream tasks. FacT [26] decomposes the weights of ViT into individual three-dimensional tensors and further decomposes the increments into lightweight factors. During fine-tuning phase, these factors are updated and stored, effectively reducing computational overhead.

## 2.3 LoRA and its variants

As represented by LoRA [1], the core of this type of PEFT method is the utilization of low-rank matrices to approximate weight adjustments during the fine-tuning phase. By employing reparameterization techniques, these low-rank matrices are combined with the existing parameter matrices, thereby circumventing extra inference costs. AdaLoRA [27] employs singular value decomposition to decompose weight matrices, pruning insignificant singular values to effectively reduce the number of parameters. ARC [3] uses symmetric up-down projections to create cross-layer shared bottleneck operations. By learning low-dimensional rescaling coefficients, it effectively recombines layer-adaptive adapters, reducing the costs of fine-tuning. FedPara [28] reparameterizes model layers with low-rank matrices and uses the Hadamard product. This approach, unconstrained by low-rank limitations, offers greater capacity and reduces learning costs. RLRR [29] examines mainstream PEFT methods from the perspective of SVD decomposition, exploring the critical balance between preserving generalization in pre-trained models and adapting them to downstream tasks. Our research abandons the traditional fixed-rank approach, opting instead for a more flexible adjustment of parameter matrices using a small number of learnable parameters.

## 3 Methodology

In this section, we commence with an introduction of the notations, symbols, and contextual background related to low-rank adaptations and Householder transformation. Then we present the decomposed structure of low-rank adaptation and discuss its inherent operating mechanism from the perspective of singular value decomposition. Finally, we propose a novel adaptation via Householder transformation. This adaptation primarily aims to construct the Householder unitary matrices via a learnable weight vector, thereby trading-off the fully spanned representation space and the affordable parameter size.

### 3.1 Preliminary knowledge

**Low-rank adaptation.** Pre-trained ViT models are typically initialized with weights learned from large-scale image datasets, such as ImageNet. The pre-training process involves optimizing the model on an unsupervised or supervised pretext task. The resulting pre-trained weights encode rich semantic information that can be transferred to a wide range of downstream tasks through fine-tuning. As one of the most representative methods of fine-tuning, PEFT method achieves excellent results on downstream tasks by merely utilizing a small number of additional learnable parameters to fine-tune the ViT. The most prevalent PEFT is the adaptation method, which can be divided into two categories: LoRA-based and adapter-based methods. In general, LoRA-based method is defined as:

$$\mathbf{X}_{\text{FT}}^{(l-1)} = \mathbf{X}^{(l-1)}(\mathbf{W}^{(l)} + \mathbf{W}_{\text{down}}^{(l)}\mathbf{W}_{\text{up}}^{(l)}) + \vec{\boldsymbol{b}}^{(l)\top}, \tag{1}$$

where $\mathbf{X}_{\text{FT}}^{(l-1)}$ is the fine-tuning output, $\mathbf{W}^{(l)}$ is any linear weight projection $\{\mathbf{W}_q^{(l)}, \mathbf{W}_k^{(l)}, \mathbf{W}_v^{(l)}, \mathbf{W}_o^{(l)}, \mathbf{W}_{\text{FC1}}^{(l)}, \mathbf{W}_{\text{FC2}}^{(l)}\}$ in ViT, $\vec{\boldsymbol{b}}^{(l)}$ is the bias weights, $\mathbf{W}_{\text{down}}^{(l)} \in \mathbb{R}^{D^{(l)} \times D'}$ and $\mathbf{W}_{\text{up}}^{(l)} \in \mathbb{R}^{D' \times D^{(l)}}$ are down- and up-adapting projection matrices across varying layers with the dimensionality $D' \ll D^{(l)}$. The detailed framework of LoRA-based method is shown in Fig. 1 (a). Analogously, adapter-based method is described as:

$$\begin{aligned} \mathbf{X}_{\text{FT}}^{(l-1)} &= \text{Act}(\text{MHA}(\mathbf{X}^{(l-1)})\mathbf{W}_{\text{down}}^{\text{MHA}(l)})\mathbf{W}_{\text{up}}^{\text{MHA}(l)}, \\ \mathbf{X}_{\text{FT}}^{(l)} &= \text{Act}(\text{FFN}(\mathbf{X}_{\text{FT}}^{(l-1)})\mathbf{W}_{\text{down}}^{\text{FFN}(l)})\mathbf{W}_{\text{up}}^{\text{FFN}(l)}, \end{aligned} \tag{2}$$

with the activation function $\text{Act}(\cdot)$, Multi-Head Attention (MHA), and Feed-Forward Network (FFN) modules in ViT. The detail of adapter-based method is shown in Fig. 1 (c). By observing

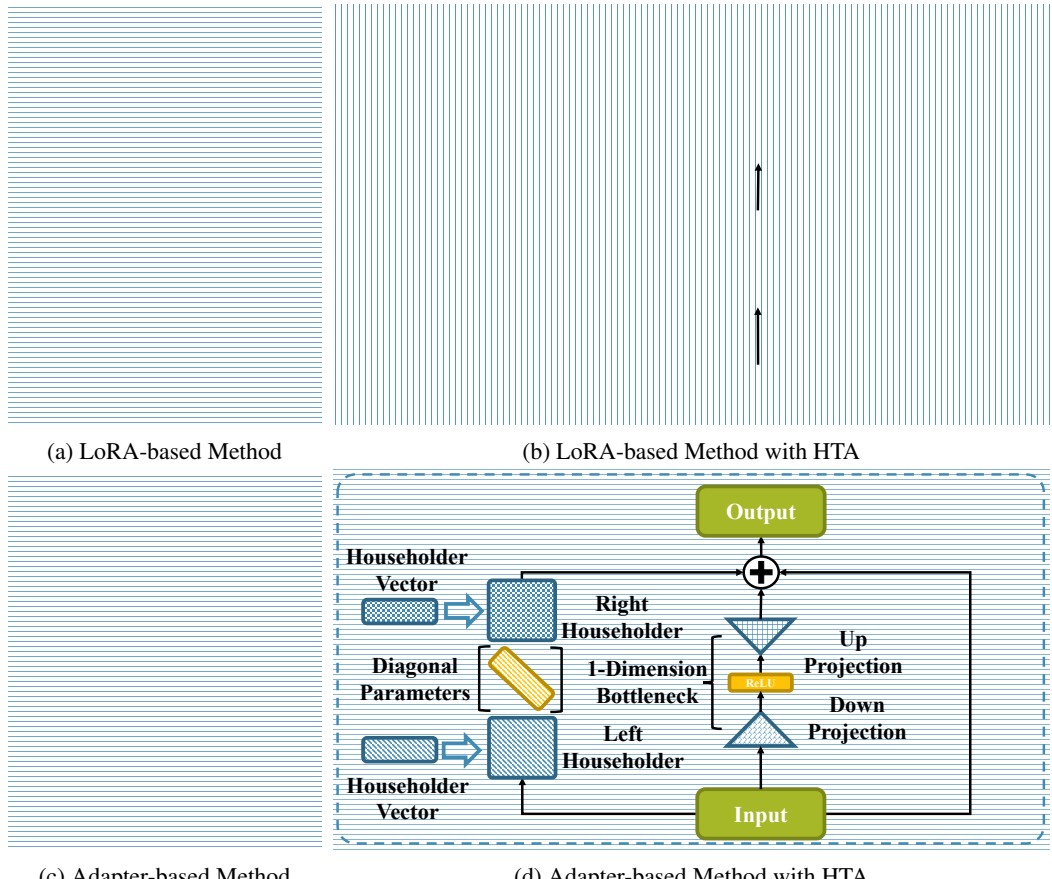

(a) LoRA-based Method      (b) LoRA-based Method with HTA

(c) Adapter-based Method      (d) Adapter-based Method with HTA

Figure 1: Underpinned by (a) LoRA [1] and (c) Adapter [22], we utilize Householder matrix to construct Householder transformation-based adaptations, involving (b) LoRA-based method with HTA and (d) Adapter-based method with HTA.

Eq. (1) and (2), both above-mentioned PEFT methods involve a low-rank bottleneck structure, *i.e.*, $\mathbf{W}^{(l)}_{\text{adaptation}} = \mathbf{W}^{(l)}_{\text{down}} \mathbf{W}^{(l)}_{\text{up}}$. Note that we remove the activation in the low-rank bottleneck because the presence or absence of such activation does not affect the low-rank structure of adaptation.

**Householder transformation.** Householder transformation, or Householder reflection, is a linear transformation that reflects a vector across a hyperplane defined by a Householder vector. It is characterized by a Householder matrix, which is an orthogonal and symmetrical matrix with determinant -1. This transformation, initially proposed by A.S. Householder in 1958 [30], has significant applications in numerical linear algebra [31], particularly in QR decomposition [32], where it is used to transform a matrix into an upper triangular or Hessenberg form [33]. Householder transformation can also be employed to set specific elements of a vector to zero while preserving its norm, making it a valuable tool for matrix orthogonalization and symmetrization. The Householder matrix is defined as following:

$$\mathbf{H} = \mathbf{I} - \vec{v}\vec{v}^{\top}, \tag{3}$$

with the identity matrix $\mathbf{I}$ and the Householder vector $\vec{v}$.

## 3.2 Viewing the adaptation matrix from SVD

Singular Value Decomposition (SVD) offers a profound insight into matrix factorization. It breaks down a matrix into three constituent matrices. Viewing the adaptation matrix through the lens of SVD, we represent it as:

$$\mathbf{W}^{(l)}_{\text{adaptation}} = \mathbf{W}^{(l)}_{\text{left}} \mathbf{D}^{(l)} \mathbf{W}^{(l)}_{\text{right}}, \tag{4}$$

where $\mathbf{W}_{\text{left}}^{(l)} \in \mathbb{R}^{D^{(l)} \times D^{(l)}}$ is the left unitary matrix and $\mathbf{W}_{\text{right}}^{(l)} \in \mathbb{R}^{D^{(l)} \times D^{(l)}}$ is the right unitary matrix; and $\mathbf{D}^{(l)} \in \mathbb{R}^{D^{(l)} \times D^{(l)}}$ is the diagonal matrix in which the diagonal elements of a diagonal matrix are singular values. Unitary matrices $\mathbf{W}_{\text{left}}^{(l)}$ and $\mathbf{W}_{\text{right}}^{(l)}$ essentially characterize the rotation transformations in a linear space. The left unitary matrix $\mathbf{W}_{\text{left}}^{(l)}$ rotates an arbitrary vector multiplied by the adaptation matrix $\mathbf{W}_{\text{adaptation}}^{(l)}$ into the space it spans. Then, the vector is scaled by the diagonal matrix $\mathbf{D}^{(l)}$. Finally, the right unitary matrix $\mathbf{W}_{\text{right}}^{(l)}$ rotates the vector back to the original linear space. Therefore, the SVD decomposition characterizes the transformations of rotation and scaling in the linear space.

When it comes to the fine-tuning strategy, PEFT methods employ ViT as the backbone and essentially fine-tune the learnable parameters of unitary matrices $\mathbf{W}_{\text{left}}^{(l)} \in \mathbb{R}^{D^{(l)} \times D'}$ and $\mathbf{W}_{\text{right}}^{(l)} \in \mathbb{R}^{D' \times D^{(l)}}$ and the learnable singular values of diagonal matrix $\mathbf{D}^{(l)} \in \mathbb{R}^{D' \times D'}$ to downstream tasks, implicitly performing the rotation and scaling transformations. Note that the number of non-zero singular values in matrix $\mathbf{D}^{(l)}$ does not exceed its dimensionality $D'$. However, the fixed bottleneck dimensionality $D'$ empirically set to LoRA- or adapter-based methods is inflexible, thereby without accommodating variations in layer-wise properties. This implies that the linear space spanned by the low-rank adaptation matrix and its corresponding number of non-zero singular values is constrained within a low dimensionality $D'$. Increasing the dimensionality $D'$ could effectively enhance the space capacity of the adaptation matrix, thereby improving the performance potential of the fine-tuned ViT model. However, this also further increases the number of parameters in the PEFT method.

### 3.3 Householder transformation-based adaptation

To address the aforementioned issue, we introduce Householder transformation into the adaptation matrix learning, and propose the Householder Transformation-based Adaptor (HTA). Following this way, HTA facilitates the derivation of the adaptation matrix in a manner that is parameter-efficient, while theoretically accommodating the flexibility of varying ranks for the adaptation matrices.

In our approach, we respectively employ the Householder matrices $\mathbf{H}_{\text{left}}^{(l)} \in \mathbb{R}^{D^{(l)} \times D^{(l)}}$ and $\mathbf{H}_{\text{right}}^{(l)} \in \mathbb{R}^{D^{(l)} \times D^{(l)}}$ as substitutes for the left and right unitary matrices $\mathbf{W}_{\text{left}}^{(l)} \in \mathbb{R}^{D^{(l)} \times D'}$ and $\mathbf{W}_{\text{right}}^{(l)} \in \mathbb{R}^{D' \times D^{(l)}}$ within the adaptation matrix $\mathbf{W}_{\text{adaptation}}^{(l)}$ to form the HTA adaptation matrix $\mathbf{W}_{\text{HTA}}^{(l)}$:

$$
\begin{aligned}
\mathbf{W}_{\text{HTA}}^{(l)} &= \mathbf{H}_{\text{left}}^{(l)} \mathbf{D}_{\text{H}}^{(l)} \mathbf{H}_{\text{right}}^{(l)} \\
&= (\mathbf{I} - \vec{v}_{\text{left}}^{(l)} \vec{v}_{\text{left}}^{(l)\top}) \mathbf{D}_{\text{H}}^{(l)} (\mathbf{I} - \vec{v}_{\text{right}}^{(l)} \vec{v}_{\text{right}}^{(l)\top})),
\end{aligned}
\tag{5}
$$

with two learnable parameter vectors $\vec{v}_{\text{left}}^{(l)} \in \mathbb{R}^{D^{(l)}}$ and $\vec{v}_{\text{right}}^{(l)} \in \mathbb{R}^{D^{(l)}}$ and a learnable diagonal parameter vector $\vec{d}^{(l)} \in \mathbb{R}^{D^{(l)}}$ in the diagonal matrix $\mathbf{D}_{\text{H}}^{(l)} \in \mathbb{R}^{D^{(l)} \times D^{(l)}}$.

Since the Householder transformation matrix is derived from a single vector, its transformation capacity can be limited and sensitive to the vector learned to derive it. To enhance the robustness of the adaptation matrix, we incorporate an additional low-rank adaptation matrix, resulting in the ultimate HTA. Building on this design, we can derive the LoRA alternative as follows:

$$
\begin{aligned}
\mathbf{X}_{\text{FT}}^{(l-1)} &= \mathbf{X}^{(l-1)} (\mathbf{W}^{(l)} + \mathbf{W}_{\text{down}}^{(l)} \mathbf{W}_{\text{up}}^{(l)} + \mathbf{W}_{\text{HTA}}^{(l)}) + \vec{b}^{(l)\top} \\
&= \mathbf{X}^{(l-1)} (\mathbf{W}^{(l)} + \mathbf{W}_{\text{down}}^{(l)} \mathbf{W}_{\text{up}}^{(l)} + (\mathbf{I} - \vec{v}_{\text{left}}^{(l)} \vec{v}_{\text{left}}^{(l)\top}) \mathbf{D}^{(l)} (\mathbf{I} - \vec{v}_{\text{right}}^{(l)} \vec{v}_{\text{right}}^{(l)\top})) + \vec{b}^{(l)\top},
\end{aligned}
\tag{6}
$$

where $\mathbf{W}_{\text{down}}^{(l)} \in \mathbb{R}^{D^{(l)} \times 1}$ and $\mathbf{W}_{\text{down}}^{(l)} \in \mathbb{R}^{1 \times D^{(l)}}$, unless otherwise stated. The HTA structure of the LoRA alternative is shown in Fig. 1 (b).

Analogously, we can derive HTA alternative to the adapter-based method (as shown in Fig. 1 (d)) as follows:

$$
\begin{aligned}
\mathbf{X}_{\text{FT}}^{(l-1)} =& \text{MHA}(\mathbf{X}^{(l-1)})(\mathbf{W}_{\text{down}}^{\text{MHA}(l)}\mathbf{W}_{\text{up}}^{\text{MHA}(l)} + \mathbf{W}_{\text{HTA}}^{\text{MHA}(l)}) \\
=& \text{MHA}(\mathbf{X}^{(l-1)})(\mathbf{W}_{\text{down}}^{\text{MHA}(l)}\mathbf{W}_{\text{up}}^{\text{MHA}(l)}+ \\
& (\mathbf{I} - \vec{\boldsymbol{v}}_{\text{left}}^{\text{MHA}(l)}\vec{\boldsymbol{v}}_{\text{left}}^{\text{MHA}(l)\top})\mathbf{D}_{\text{H}}^{\text{MHA}(l)}(\mathbf{I} - \vec{\boldsymbol{v}}_{\text{right}}^{\text{MHA}(l)}\vec{\boldsymbol{v}}_{\text{right}}^{\text{MHA}(l)\top})), \\
\mathbf{X}_{\text{FT}}^{(l)} =& \text{FFN}(\mathbf{X}_{\text{FT}}^{(l-1)})(\mathbf{W}_{\text{down}}^{\text{FFN}(l)}\mathbf{W}_{\text{up}}^{\text{FFN}(l)} + \mathbf{W}_{\text{HTA}}^{\text{FFN}(l)}) \\
=& \text{FFN}(\mathbf{X}_{\text{FT}}^{(l-1)})(\mathbf{W}_{\text{down}}^{\text{FFN}(l)}\mathbf{W}_{\text{up}}^{\text{FFN}(l)}+ \\
& (\mathbf{I} - \vec{\boldsymbol{v}}_{\text{left}}^{\text{FFN}(l)}\vec{\boldsymbol{v}}_{\text{left}}^{\text{FFN}(l)\top})\mathbf{D}_{\text{H}}^{\text{FFN}(l)}(\mathbf{I} - \vec{\boldsymbol{v}}_{\text{right}}^{\text{FFN}(l)}\vec{\boldsymbol{v}}_{\text{right}}^{\text{FFN}(l)\top})).
\end{aligned}
\tag{7}
$$

By observing Eq. (6), we can see that LoRA-based method with HTA could be re-parameterized to the form $\mathbf{W}_{\text{re}-\text{param}} = \mathbf{W}_{\text{down}}^{(l)}\mathbf{W}_{\text{up}}^{(l)} + \mathbf{W}_{\text{HTA}}^{(l)}$ during the model inference stage. And also, the re-parameterization $\mathbf{W}_{\text{re}-\text{param}}^{\text{MHA}} = \mathbf{W}_{o}^{(l)}\mathbf{W}_{\text{HTA}}^{(l)}$ of MHA in Eq. (7) is available due to the fact that the weight matrix $\mathbf{W}_{o}^{(l)}$ is positioned at the end of MHA. Similarly, the re-parameterization of FFN is $\mathbf{W}_{\text{re}-\text{param}}^{\text{FFN}} = \mathbf{W}_{\text{FC2}}^{(l)}\mathbf{W}_{\text{HTA}}^{(l)}$ due to the weight matrix $\mathbf{W}_{\text{FC2}}^{(l)}$ at the tail of FFN.

# 4  Experiments

In this section, we present the experimental settings, comparison to existing solutions, and ablation studies to unveil the key properties of the proposed method.

## 4.1  Experimental settings

**Datasets.** We evaluated the effectiveness of our method using two sets of visual adaptation benchmarks: FGVC and VTAB-1k, involving a total of 24 datasets. The FGVC collection consists of five Fine-Grained Visual Classification (FGVC) datasets: CUB-200-2011, NABirds, Oxford Flowers, Stanford Dogs, and Stanford Cars. These datasets focus on distinguishing between visually similar subcategories within a broader category, making the task more challenging and detailed. The VTAB-1k benchmark comprises 19 diverse visual classification tasks, divided into three categories: Natural, which includes images captured by standard cameras; Specialized, which includes images captured by specialized equipment such as remote sensing and medical imaging devices; and Structured, which includes synthesized images from simulated environments, such as object counting and 3D depth prediction. Each VTAB-1k task includes 1,000 training samples.

**Pre-trained backbones.** We employ ViT [13] and Swin Transformer [14] as backbone architectures to evaluate our proposed approach. To demonstrate the versatility of the proposed HTA model, we utilize two variants of ViT: ViT-Base and ViT-Large. These models are pre-trained on the ImageNet21K dataset [11]. Additionally, to ensure a fair comparison, we follow the settings from previous work [29] and conduct separate experiments using a ViT backbone enhanced with AugReg [34].

**Baselines and existing PEFT methods.** In our comparative analysis, we evaluate the performance of our HTA against two baselines and several state-of-the-art PEFT methods. Unless otherwise specified, our HTA follows the design in Eq. (6), with the dimension of the low-rank adaptation matrix set to 1. The two baselines we consider are: (1) Full Fine-tuning: This baseline involves updating all the parameters of the pre-trained model using the training data of the downstream task. (2) Linear Probing: This baseline focuses on learning a linear classification head on the downstream task while keeping the remaining parameters of the pre-trained model frozen. In addition to the baselines, we compare our method with the following state-of-the-art solutions: Adapter [22], Bias [23], LoRA [1], VPT [24], AdaptFormer [2], FacT [26], ARC [3] and RLRR [29]. The results are presented in Table 1 and Table 2.

Table 1: Performance comparisons on the VTAB-1k benchmark with ViT-B/16 models pre-trained on ImageNet-21K. * denotes leveraging the augmented ViT backbone by AugReg [34]. The **bold** font shows the best accuracy of all methods and the underline font shows the second best accuracy.

| Methods / Datasets | CIFAR-100 | Caltech101 | DTD | Flowers102 | Pets | SVNH | Sun397 | Mean | Camelyon | EuroSAT | Resisc45 | Retinopathy | Mean | Clevr-Count | Clevr-Dist | DMLab | KITTI-Dist | dSpr-Loc | dSpr-Ori | sNORB-Azim | sNORB-Ele | Mean | Mean Total | Params.(M) |
|---|---|---|---|---|---|---|---|---|---|---|---|---|---|---|---|---|---|---|---|---|---|---|---|---|
| | | | | Natural | | | | | | | Specialized | | | | | | | Structed | | | | | | |
| Full fine-tuning | 68.9 | 87.7 | 64.3 | 97.2 | 86.9 | 87.4 | 38.8 | 75.9 | 79.7 | 95.7 | 84.2 | 73.9 | 83.4 | 56.3 | 58.6 | 41.7 | 65.5 | 57.5 | 46.7 | 25.7 | 29.1 | 47.6 | 65.6 | 85.80 |
| Linear probing | 63.4 | 85.0 | 63.2 | 97.0 | 86.3 | 36.6 | 51.0 | 68.9 | 78.5 | 87.5 | 68.6 | 74.0 | 77.2 | 34.3 | 30.6 | 33.2 | 55.4 | 12.5 | 20.0 | 9.6 | 19.2 | 26.9 | 52.9 | 0.04 |
| Bias [23] | 72.8 | 87.0 | 59.2 | 97.5 | 85.3 | 59.9 | 51.4 | 73.3 | 78.7 | 91.6 | 72.9 | 69.8 | 78.3 | 61.5 | 55.6 | 32.4 | 55.9 | 66.6 | 40.0 | 15.7 | 25.1 | 44.1 | 62.1 | 0.14 |
| VPT-Shallow [24] | 77.7 | 86.9 | 62.6 | 97.5 | 87.3 | 74.5 | 51.2 | 76.8 | 78.2 | 92.0 | 75.6 | 72.9 | 79.7 | 50.5 | 58.6 | 40.5 | 67.1 | 68.7 | 36.1 | 20.2 | 34.1 | 47.0 | 64.9 | 0.11 |
| VPT-Deep [24] | **78.8** | 90.8 | 65.8 | 98.0 | 88.3 | 78.1 | 49.6 | 78.5 | 81.8 | **96.1** | 83.4 | 68.4 | 82.4 | 68.5 | 60.0 | 46.5 | 72.8 | 73.6 | 47.9 | 32.9 | 37.8 | 55.0 | 69.4 | 0.60 |
| Adapter [22] | 69.2 | 90.1 | 68.0 | 98.8 | 89.9 | 82.8 | 54.3 | 79.0 | 84.0 | 94.9 | 81.9 | 75.5 | 84.1 | 80.9 | 65.3 | 48.6 | 78.3 | 74.8 | 48.5 | 29.9 | 41.6 | 58.5 | 71.4 | 0.16 |
| LORA [1] | 67.1 | 91.4 | 69.4 | 98.8 | 90.4 | 85.3 | 54.0 | 79.5 | 84.9 | 95.3 | 84.4 | 73.6 | 84.6 | **82.9** | **69.2** | 49.8 | 78.5 | 75.7 | 47.1 | 31.0 | 44.0 | 59.8 | 72.3 | 0.29 |
| AdaptFormer [2] | 70.8 | 91.2 | 70.5 | 99.1 | 90.9 | 86.6 | 54.8 | 80.6 | 83.0 | 95.8 | 84.4 | 76.3 | 84.9 | 81.9 | 64.3 | 49.3 | 80.3 | 76.3 | 45.7 | 31.7 | 41.1 | 58.8 | 72.3 | 0.16 |
| FacT-TK$_{<32}$[26] | 70.6 | 90.6 | 70.8 | 99.1 | 90.7 | 88.6 | 54.1 | 80.6 | 84.8 | 96.2 | 84.5 | 75.7 | 85.3 | 82.6 | 68.2 | 49.8 | 80.7 | 80.8 | 47.4 | 33.2 | 43.0 | 60.7 | 73.2 | 0.07 |
| ARC [3] | 72.2 | 90.1 | 72.7 | 99.0 | 91.0 | **91.9** | 54.4 | 81.6 | 84.9 | 95.7 | **86.7** | 75.8 | 85.8 | 80.7 | 67.1 | 48.7 | 81.6 | 79.2 | 51.0 | 31.4 | 39.9 | 60.0 | 73.4 | 0.13 |
| RLRR [29] | 75.6 | 92.4 | **72.9** | **99.3** | **91.5** | 89.8 | **57.0** | **82.7** | 86.8 | 95.2 | 85.3 | 75.9 | 85.8 | 79.7 | 64.2 | **53.9** | **82.1** | 83.9 | **53.7** | 33.4 | 43.6 | 61.8 | 74.5 | 0.33 |
| HTA | 76.6 | **94.3** | 72.5 | **99.3** | 91.3 | 86.2 | 56.5 | 82.4 | **87.6** | 95.7 | 85.0 | 75.7 | **86.0** | 82.6 | 63.3 | 52.5 | 81.0 | **84.5** | 52.6 | **34.5** | **47.3** | **62.3** | **74.7** | 0.22 |
| SSF [25] | 69.0 | 92.6 | 75.1 | 99.4 | 91.8 | 90.2 | 52.9 | 81.6 | 87.4 | 95.9 | 87.4 | 75.5 | 86.6 | 75.9 | 62.3 | 53.3 | 80.6 | 77.3 | 54.9 | 29.5 | 37.9 | 59.0 | 73.1 | 0.24 |
| ARC* [3] | 71.2 | 90.9 | 75.9 | 99.5 | 92.1 | 90.8 | 52.0 | 81.8 | 87.4 | **96.5** | 87.6 | 76.4 | 87.0 | 83.3 | 61.1 | 54.6 | 81.7 | 81.0 | **57.0** | 30.9 | 41.3 | 61.4 | 74.3 | 0.13 |
| RLRR* [29] | 76.7 | 92.7 | 76.3 | **99.6** | **92.6** | **91.8** | 56.0 | **83.7** | 87.8 | 96.2 | 89.1 | 76.3 | 87.3 | 80.4 | 63.3 | 54.5 | **83.3** | 83.0 | 53.7 | 32.0 | 41.7 | 61.5 | 75.1 | 0.33 |
| HTA* | **79.0** | 92.8 | **77.6** | **99.6** | 92.4 | 89.4 | 55.1 | **83.7** | **88.2** | 96.1 | **89.7** | 76.4 | **87.6** | **84.2** | 61.7 | 53.6 | 82.0 | **85.1** | 53.7 | 33.9 | 47.9 | **62.8** | **75.7** | 0.22 |

Table 2: Performance comparisons on five FGVC datasets with ViT-B/16 models pre-trained on ImageNet-21K. * denotes leveraging the augmented ViT backbone by AugReg [34].

| Methods / Datasets | CUB-200-2011 | NABirds | Oxford Flowers | Stanford Dogs | Stanford Cars | Mean Total | Params. (M) |
|---|---|---|---|---|---|---|---|
| Full fine-tuning | 87.3 | 82.7 | 98.8 | 89.4 | 84.5 | 88.5 | 85.98 |
| Linear probing | 85.3 | 75.9 | 97.9 | 86.2 | 51.3 | 79.3 | 0.18 |
| Adapter [22] | 87.1 | 84.3 | 98.5 | 89.8 | 68.6 | 85.7 | 0.41 |
| Bias [23] | 88.4 | 84.2 | 98.8 | 91.2 | 79.4 | 88.4 | 0.28 |
| VPT-Shallow [24] | 86.7 | 78.8 | 98.4 | 90.7 | 68.7 | 84.6 | 0.25 |
| VPT-Deep [24] | 88.5 | 84.2 | 99.0 | 90.2 | 83.6 | 89.1 | 0.85 |
| LoRA [1] | 88.3 | **85.6** | 99.2 | 91.0 | 83.2 | 89.5 | 0.44 |
| ARC [3] | 88.5 | 85.3 | 99.3 | 91.9 | 85.7 | 90.1 | 0.25 |
| RLRR [29] | **89.3** | 84.7 | **99.5** | 92.0 | 87.0 | 90.4 | 0.47 |
| HTA | 88.8 | 84.4 | **99.5** | **92.2** | **87.9** | **90.6** | 0.36 |
| SSF [25] | 89.5 | **85.7** | 99.6 | 89.6 | 89.2 | 90.7 | 0.39 |
| ARC* [3] | 89.3 | **85.7** | **99.7** | 89.1 | 89.5 | 90.7 | 0.25 |
| RLRR* [29] | 89.8 | 85.3 | 99.6 | **90.0** | 90.4 | 91.0 | 0.47 |
| HTA* | **90.5** | 85.4 | 99.6 | 89.3 | **90.5** | **91.1** | 0.36 |

**Implementation details.** Following previous work, we employed data augmentation during the training phase. For the FGVC datasets, we processed the images with a random resize crop to $224 \times 224$ and applied a random horizontal flip for data augmentation. For the VTAB-1k datasets, we directly resized the images to $224 \times 224$, adhering to the default settings in VTAB-1k. We used the AdamW [35] optimizer to fine-tune the models for 100 epochs. The learning rate schedule was managed using the cosine decay strategy. All experiments are conducted using the PyTorch framework [36] on an NVIDIA A800 GPU with 80 GB of memory.

## 4.2 Experimental comparisons

In this section, we conduct a comprehensive comparison of our method with other state-of-the-art approaches using different benchmarks and backbones. We evaluate the classification accuracy of each method across various downstream tasks and examine the number of trainable parameters during the fine-tuning phase.

**Comparison with the existing PEFT methods.** We conducted a comparison of our method with other PEFT methods and baselines using two different benchmarks: FGVC and VTAB-1k. The results are presented in Table 1 and Table 2. On the VTAB-1k dataset, our method not only demonstrates strong competitiveness compared to the baselines but also shows advantages over current state-of-the-art methods. On many of the datasets, our method achieves the best performance with a reasonable parameter count. Compared to the previous state-of-the-art method, RLRR [29], our method achieves superior overall performance while reducing the number of parameters by one-third. When using the AugReg-enhanced model, our lead is further amplified. In Table 2, we further compare our method with others on the FGVC benchmark. While our method also achieves appealing performance on this dataset, the advantage is less evident. This is due to the fact that very high performance has already been achieved on this dataset, and the performance improvements have nearly saturated.

Table 3: Performance comparison on VTAB-1k using ViT-Large pre-trained on ImageNet-21k as the backbone. Detailed results are presented in the Appendix.

| Datasets
Methods | Natural (7) | Specialized (4) | Structed (8) | Mean Total | Params.(M) |
|---|---|---|---|---|---|
| Full fine-tuning | 74.7 | 83.8 | 48.1 | 65.4 | 303.40 |
| Linear probing | 70.9 | 69.1 | 25.8 | 51.5 | 0.05 |
| Bias [23] | 70.5 | 73.8 | 41.2 | 58.9 | 0.32 |
| VPT-Shallow [24] | 78.7 | 79.9 | 40.6 | 62.9 | 0.15 |
| VPT-Deep [24] | 82.5 | 83.9 | 54.1 | 70.8 | 0.49 |
| LoRA [1] | 81.4 | 85.0 | 57.3 | 72.0 | 0.74 |
| SSF [25] | 81.9 | 85.2 | 59.0 | 73.0 | 0.60 |
| ARC [3] | 82.3 | 85.6 | 57.3 | 72.5 | 0.18 |
| RLRR [29] | 83.9 | 86.4 | 61.9 | 75.2 | 0.82 |
| HTA | **84.1** | **86.6** | **62.3** | **75.4** | 0.54 |

Table 4: Performance comparison on VTAB-1k using Swin Transformer pre-trained on ImageNet-21k as the backbone. Detailed results are presented in the Appendix.

| Datasets
Methods | Natural (7) | Specialized (4) | Structed (8) | Mean Total | Params.(M) |
|---|---|---|---|---|---|
| Full fine-tuning | 79.1 | 86.2 | 59.7 | 72.4 | 86.80 |
| Linear probing | 73.5 | 80.8 | 33.5 | 58.2 | 0.05 |
| MLP-4 [24] | 70.6 | 80.7 | 31.2 | 57.7 | 4.04 |
| Partial [24] | 73.1 | 81.7 | 35.0 | 58.9 | 12.65 |
| Bias [23] | 74.2 | 80.1 | 42.4 | 62.1 | 0.25 |
| VPT-Shallow [24] | 79.9 | 82.5 | 37.8 | 62.9 | 0.05 |
| VPT-Deep [24] | 76.8 | 84.5 | 53.4 | 67.7 | 0.22 |
| ARC [3] | 79.0 | 86.6 | 59.9 | 72.6 | 0.27 |
| RLRR [29] | 81.3 | **86.7** | 59.0 | 73.0 | 0.41 |
| HTA | **81.8** | **86.7** | **61.3** | **74.2** | 0.23 |

**Experiments on larger-scale ViT backbone.** In addition to using the ViT-B backbone, we also employed the ViT-L backbone, which has a deeper block structure, to validate the scalability and generalizability of our method. The comparison results are shown in Table 3. Our method achieves the best performance among all the compared methods while maintaining a reasonable parameter count. These results demonstrate that our method can effectively adapt models of varying scales and complexities in an efficient manner.

**Experiments on hierarchical Vision Transformers.** To further validate the effectiveness of our method, we tested it on the Swin Transformer [14]. The Swin Transformer is renowned for its hierarchical design, consisting of multiple stages, each with transformer blocks that maintain consistent feature dimensions unique to that stage. As shown in Table 4, our method notably outperforms existing state-of-the-art methods across various downstream tasks, with an overall improvement of 1.2% over the previous best performance while using only half of the parameters.

### 4.3 Ablation studies

To gain deeper insights into the proposed method, we conducted comprehensive ablation studies to elucidate its critical features and carry out pertinent analyses.

**Using HTA as alternative to low-rank based adaptation matrix.** As mentioned earlier, our proposed HTA model offers a more flexible adaptation capacity compared to other low-rank based adaptation matrices. To validate this claim, we replaced the adaptation matrices of LoRA [1] and Adaptor [22] with HTA. Initially, following FacT [26], LoRA [1] was originally applied to the $\{\mathbf{W}_q, \mathbf{W}_v\}$ projection matrices in the multi-head attention operation of each ViT layer, while Adapter was applied to the feed-forward neural network components layer-wise, as described in Eq. (2). For

Table 5: Ablation study on using HTA as alternative to the low-rank based adaptation matrices in LoRA and Adapter on VTAB-1k. Following the configurations in FacT [26], LoRA and Adapter are applied to $\{\mathbf{W}_q, \mathbf{W}_v\}$ and $\{\mathbf{W}_{FC1}, \mathbf{W}_{FC2}\}$ projection matrices, separately.

| Datasets / Methods | Natural (7) | Specialized (4) | Structed (8) | Mean Total | Params.(M) |
|---|---|---|---|---|---|
| LoRA ($\mathbf{W}_q$, $\mathbf{W}_v$) | 79.5 | 84.6 | 59.8 | 72.3 | 0.29 |
| HTA ($\mathbf{W}_q$, $\mathbf{W}_v$) | 81.0 | 84.6 | 59.6 | 72.7 | 0.09 |
| HTA ($\mathbf{W}_q$, $\mathbf{W}_v$, $\mathbf{W}_{FC1}$, $\mathbf{W}_{FC2}$) | 81.1 | 86.3 | 61.5 | 73.9 | 0.28 |
| Adapter ($\mathbf{W}_{FC1}$, $\mathbf{W}_{FC2}$) | 79.0 | 84.1 | 58.5 | 71.4 | 0.16 |
| HTA ($\mathbf{W}_{FC1}$, $\mathbf{W}_{FC2}$) | 81.0 | 84.9 | 60.0 | 73.0 | 0.05 |

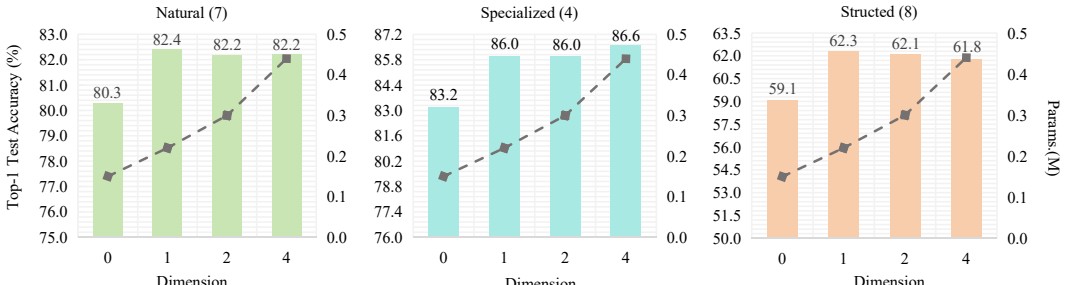

Figure 2: Ablation study on the impact of different bottleneck dimensions of adaptive matrices in HTA. The bar chart represents the Top-1 Test Accuracy, the line graph indicates parameters count.

a fair comparison, we also applied HTA separately to $\{\mathbf{W}_q, \mathbf{W}_v\}$ and $\{\mathbf{W}_{FC1}, \mathbf{W}_{FC2}\}$. From the results in Table 5, we observe that our method slightly outperforms LoRA but with significantly fewer parameters. Moreover, our method achieves significant improvement over Adapter still using much fewer parameters. These results indicate that our method achieves a better trade-off between adaptation performance and parameter efficiency. To further test the effectiveness of our method when using a similar parameter scale to LoRA, we additionally applied HTA to FFN layers. The results show that under the same parameter size, our method exhibits a noticeable improvement over LoRA.

**Ablation study on the bottleneck dimensionality of additive adaptation matrix in HTA.** We conducted ablation experiments to verify the effect of incorporating low-rank adaptation matrices in HTA, as well as the impact of its bottleneck dimensionality. The results are presented in Fig. 2. From these results, we observe that without the addition of low-rank adaptation, HTA experiences an obvious performance drop. This is due to the fact that while deriving orthogonal matrices using Householder transformations is parameter-efficient, their inherent dependence on a single chosen vector makes them insufficient as a set of general orthogonal bases for representing arbitrary high-dimensional space. When using a low-rank adaptation matrix with rank 1, HTA shows a significant performance boost. This indicates that even with a simple low-rank adaptation, HTA can achieve a promising trade-off between adaptation performance and parameter efficiency. By incorporating these low-rank matrices, HTA can maintain high performance while being parameter-efficient.

## 5 Limitations

In this work, we use Householder transformations to construct adaptation matrices in a parameter-efficient manner. Although Householder transformation matrices are orthogonal, they cannot serve as general orthogonal bases in high-dimensional spaces due to their inherent dependence on a single vector. This limitation may reduce the adaptation capacity of the adaptation matrix composed of two Householder matrices. We address this issue by incorporating a rank-1 adaptation matrix, which may somewhat detract from the elegance of the proposed method. However, it is worth exploring strategies to eliminate the need for the additive adaptation matrix, thereby further enhancing the elegance and efficiency of the HTA method.

# 6    Conclusions

In this work, we proposed a novel Parameter-Efficient Fine-Tuning (PEFT) solution. Our method addresses the limitation of fixed bottleneck dimensionality in low-rank based adaptation matrices, which can restrict adaptation flexibility. By viewing the adaptation matrix from the perspective of Singular Value Decomposition (SVD), we use Householder transformations to mimic orthogonal bases. These transformations, derived from a single vector, are highly parameter-efficient. We adaptively learn diagonal coefficients to flexibly combine two Householder matrices into an adaptation matrix, accommodating layer-wise variations. Theoretically, our method can generate adaptation matrices with varying ranks while maintaining a reasonable parameter size, offering a potential alternative to low-rank based adaptations. Experiments on two sets of downstream vision classification tasks demonstrate the effectiveness of our approach.

## Acknowledgments and Disclosure of Funding

W. Dong's participation was in part supported by the Natural Science Basic Research Program of Shaanxi (Program No.2024JC-YBMS-464), Major Science and Technology Innovation Project of Xianyang (Program No.L2024-ZDKJ-ZDGG-GY-0010).

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

# Efficient Adaptation of Pre-trained Vision Transformer via Householder Transformation

## Supplementary Materials

In the supplementary materials involving our work, we demonstrate Detailed dataset statistic, Hyper-parameters in our work, Experimental details, and broader impacts, including:

- A **Detailed dataset statistic**

- B **Hyper-parameters in our work**

- C **Experimental details on larger-scale and hierarchical ViT backbones**

- D **Experimental details on ablation study**

- E **Broader impacts**

Due to the limitation that supplementary materials larger than 100MB cannot be uploaded to the Open-Review website, only the project code as the concise supplementary materials is uploaded to this website. Please reler to the anonymous link`https://drive.google.com/file/d/18sXhtqMlKZd4_LRICk2NvSlKiFiHrG2d/view?`to obtain the complete code, datasets, and models.

# A  Detailed dataset statistic

We provide detailed information about the datasets used in this paper, including the number of classes and the sizes of the training, validation, and test sets, in Table 1 (FGVC) and Table 2 (VTAB-1k).The FGVC datasets include CUB-200-2011, NABirds, Oxford Flowers, Stanford Dogs, and Stanford Cars, which are used for fine-grained classification tasks of birds, flowers, dogs, and cars, respectively. The VTAB-1k datasets cover natural, specialized, and structured tasks, including natural image datasets such as CIFAR-100, Caltech101, DTD, Flowers102, Pets, SVHN, and Sun397; specialized image datasets such as Patch Camelyon, EuroSAT, Resisc45, and Retinopathy; and structured image datasets such as Clevr/count, Clevr/distance, DMLab, KITTI/distance, dSprites/location, dSprites/orientation, SmallNORB/azimuth, and SmallNORB/elevation. Detailed information about these datasets is presented in the tables.

Table 1: Dataset statistics for FGVC. "*" denotes the train/val split of datasets following the dataset setting in VPT [24].

| Dataset | Description | Classes | Train size | Val size | Test size |
|---------|-------------|---------|-----------|----------|-----------|
| CUB-200-2011 [37] | Fine-grained bird species recognition | 200 | 5,394* | 600* | 5,794 |
| NABirds [38] | Fine-grained bird species recognition | 555 | 21,536* | 2,393* | 24,633 |
| Oxford Flowers [39] | Fine-grained flower species recognition | 102 | 1,020 | 1,020 | 6,149 |
| Stanford Dogs [40] | Fine-grained dog species recognition | 120 | 10,800* | 1,200* | 8,580 |
| Stanford Cars [41] | Fine-grained car classificatio | 196 | 7,329* | 815* | 8,041 |

# B  Hyper-parameters in our work

Table 3 provides a summary of the configurations used in our experiments. As discussed in Section 4, we performed a grid search on the validation set of each task to determine the optimal hyperparameters, including learning rate, weight decay, batch size, and dropout rate.

# C  Experimental details on larger-scale and hierarchical ViT backbones

Table 4 and 5 respectively display the comprehensive results of the comparison conducted in Section 4 among ViT-Large and Swin-Base models.

# D  Experimental details on ablation study

We provide further explanation of the ablation experiments in Section 4. In the study on the transferability of HTA, we replaced the low-rank adaptation matrices in LoRA, we used an HTA module to replace the bottleneck part of LoRA, while in Adapter, we directly replaced the Adapter with an HTA module. The detailed experimental results are presented in the table 6. In the study of the low-rank adaptation part of HTA, we set its dimensions to 0, 1, 2, and 4, respectively. The results are shown in Table 7.

# E  Broader impacts

Practicality: Our approach differs from traditional methods by employing Householder transformations rather than standard unitary matrices, which can be efficiently derived. This approach boosts the efficiency of parameter usage and significantly cuts down on the number of parameters requiring fine-tuning. With this technique, we manage to achieve high performance while optimizing parameter use. Leveraging large-scale pre-trained models, our HTA method proves to be both highly efficient and practical across diverse applications.

Low Energy Consumption: Our approach enhances the model's computational efficiency by decreasing the necessary computational parameters, thus reducing energy usage during training. This reduction aids in conserving energy and lowering emissions, aligning with global sustainability goals and the push for eco-friendly practices. Moreover, our method not only improves the model's

Table 2: Dataset statistics for VTAB-1k [42].

| Dataset | Description | Classes | Train size | Val size | Test size |
|---|---|---|---|---|---|
| CIFAR-100 | | 100 | | | 10,000 |
| Caltech101 | | 102 | | | 6,084 |
| DTD | | 47 | | | 1,880 |
| Flowers102 | Natural | 102 | 800/1,000 | 200 | 6,149 |
| Pets | | 37 | | | 3,669 |
| SVHN | | 10 | | | 26,032 |
| Sun397 | | 397 | | | 21,750 |
| Patch Camelyon | | 2 | | | 32,768 |
| EuroSAT | Specialized | 10 | 800/1,000 | 200 | 5,400 |
| Resisc45 | | 45 | | | 6,300 |
| Retinopathy | | 5 | | | 42,670 |
| Clevr/count | | 8 | | | 15,000 |
| Clevr/distance | | 6 | | | 15,000 |
| DMLab | | 6 | | | 22,735 |
| KITTI/distance | | 4 | | | 711 |
| dSprites/location | Structured | 16 | 800/1,000 | 200 | 73,728 |
| dSprites/orientation | | 16 | | | 73,728 |
| SmallNORB/azimuth | | 18 | | | 12,150 |
| SmallNORB/elevation | | 9 | | | 12,150 |

Table 3: The implementation details of configurations such as optimizer and hyper-parameters. We select the best hyper-parameters for each download task via using grid search.

| Optimizer | AdamW |
|---|---|
| Learning Rate | {0.2, 0.1, 0.05, 0.01, 0.005, 0.001, 0.0001} |
| Weight Decay | {0.05, 0.01, 0.005, 0.001, 0} |
| Batch Size | {64, 32, 16} |
| Adapter Dropout | {0.5, 0.3, 0.2, 0.1, 0} |
| Learning Rate Schedule | Cosine Decay |
| Training Epochs | 100 |
| Warmup Epochs | 10 |

Table 4: This table is extended from Table 3 in Section 4 and describes the detailed experimental results of the performance comparison on VTAB-1k using ViT-Large pre-trained on ImageNet-21k as the backbone.

| Methods | CIFAR-100 | Caltech101 | DTD | Flowers102 | Pets | SVHN | Sun397 | Mean | Camelyon | EuroSAT | Resisc45 | Retinopathy | Mean | Clevr-Count | Clevr-Dist | DMLab | KITTI-Dist | dSpr-Loc | dSpr-Ori | sNORB-Azim | sNORB-Ele | Mean | Mean Total | Params.(M) |
|---|---|---|---|---|---|---|---|---|---|---|---|---|---|---|---|---|---|---|---|---|---|---|---|---|
| | | | | Natural | | | | | | | Specialized | | | | | | | Structed | | | | | | |
| Full fine-tuning | 68.6 | 84.3 | 58.6 | 96.3 | 86.5 | 87.5 | 41.4 | 74.7 | 82.6 | 95.9 | 82.4 | 74.2 | 83.8 | 55.4 | 55.0 | 42.2 | 74.2 | 56.8 | 43.0 | 28.5 | 29.7 | 48.1 | 65.4 | 303.4 |
| Linear probing | 72.2 | 86.4 | 63.6 | 97.4 | 85.8 | 38.1 | 52.5 | 70.9 | 76.9 | 87.3 | 66.6 | 45.4 | 69.1 | 28.2 | 28.0 | 34.7 | 54.0 | 10.6 | 14.2 | 14.6 | 21.9 | 25.8 | 51.5 | 0.05 |
| Adapter [22] | 75.3 | 84.2 | 54.5 | 97.4 | 84.3 | 31.3 | 52.9 | 68.6 | 75.8 | 85.1 | 63.4 | 69.5 | 73.5 | 35.4 | 34.1 | 30.8 | 47.1 | 30.4 | 23.4 | 10.8 | 19.8 | 29.0 | 52.9 | 2.38 |
| Bias [23] | 71.0 | 82.4 | 51.3 | 96.3 | 83.2 | 59.5 | 49.9 | 70.5 | 72.9 | 87.9 | 63.1 | 71.3 | 73.8 | 51.2 | 50.7 | 33.5 | 54.8 | 65.9 | 37.3 | 13.7 | 22.2 | 41.2 | 58.9 | 0.32 |
| VPT-Shallow [24] | 80.6 | 88.2 | 67.1 | 98.0 | 85.9 | 78.4 | 53.0 | 78.7 | 79.7 | 93.5 | 73.4 | 73.1 | 79.9 | 41.5 | 52.5 | 32.3 | 64.2 | 48.3 | 35.3 | 21.6 | 28.8 | 40.6 | 62.9 | 0.15 |
| VPT-Deep [24] | **84.1** | 88.9 | 70.8 | 98.8 | 90.0 | 89.0 | 55.9 | 82.5 | 82.5 | **96.6** | 82.6 | 73.9 | 83.9 | 63.7 | 60.7 | 46.1 | 75.7 | 83.7 | 47.4 | 18.9 | 36.9 | 54.1 | 70.8 | 0.49 |
| LoRA [1] | 75.8 | 89.8 | 73.6 | 99.1 | 90.8 | 83.2 | 57.5 | 81.4 | 86.0 | 95.0 | 83.4 | 75.5 | 85.0 | 78.1 | 60.5 | 46.7 | **81.6** | 76.7 | 51.3 | 28.0 | 35.4 | 57.3 | 72.0 | 0.74 |
| ARC [3] | 76.2 | 89.6 | 73.4 | 99.1 | 90.3 | **90.9** | 56.5 | 82.3 | 85.0 | 95.7 | 85.9 | 75.8 | 85.6 | 78.6 | 62.1 | 46.7 | 76.7 | 75.9 | 53.0 | 30.2 | 35.2 | 57.3 | 72.5 | 0.18 |
| SSF [25] | 73.5 | 91.3 | 70.0 | 99.3 | 91.3 | 90.6 | 57.5 | 81.9 | 85.9 | 94.9 | 85.5 | 74.4 | 85.2 | 80.6 | 60.0 | 53.3 | 80.0 | 77.6 | 54.0 | 31.8 | 35.0 | 59.0 | 73.0 | 0.60 |
| RLRR [29] | 79.3 | 92.0 | 74.6 | **99.5** | 92.1 | 89.6 | **60.1** | 83.9 | 87.3 | 95.3 | **87.3** | 75.7 | 86.4 | **82.7** | 62.1 | **54.6** | 80.6 | 87.1 | 54.7 | 31.3 | 41.9 | 61.9 | 75.2 | 0.82 |
| HTA | 80.8 | **92.4** | **76.1** | **99.5** | **92.8** | 87.2 | 59.9 | **84.1** | **87.7** | 95.5 | 86.8 | 76.5 | **86.6** | 82.6 | **62.4** | 53.4 | 80.0 | **87.1** | 53.7 | **33.4** | 45.6 | **62.3** | **75.4** | 0.54 |

performance and efficiency but also promotes environmental sustainability by embracing sustainable development principles.

Ethical Aspects: Our model utilizes the vast capabilities of large-scale pre-trained models for representation and generalization. However, it is trained on datasets that might contain problematic data, such as illegal content or inherent biases, which our model could inadvertently learn. To tackle this challenge, addressing model toxicity becomes critical. Consequently, it's imperative to develop

Table 5: This table is extended from Table 4 in Section 4 and describes the detailed experimental results of the performance comparison on VTAB-1k using Swin-Base pre-trained on ImageNet-21k as the backbone.

| Methods | Natural | | | | | | | | Specialized | | | | | Structed | | | | | | | | | Mean Total | Params.(M) |
|---|---|---|---|---|---|---|---|---|---|---|---|---|---|---|---|---|---|---|---|---|---|---|---|---|
| | CIFAR-100 | Caltech101 | DTD | Flowers102 | Pets | SVNH | Sun397 | Mean | Camelyon | EuroSAT | Resisc45 | Retinopathy | Mean | Clevr-Count | Clevr-Dist | DMLab | KITTI-Dist | dSpr-Loc | dSpr-Ori | sNORB-Azim | sNORB-Ele | Mean | | |
| Full fine-tuning | 72.2 | 88.0 | 71.4 | 98.3 | 89.5 | 89.4 | 45.1 | 79.1 | 86.6 | **96.9** | **87.7** | 73.6 | 86.2 | 75.7 | 59.8 | 54.6 | 78.6 | 79.4 | 53.6 | **34.6** | **40.9** | 59.7 | 72.4 | 86.9 |
| Linear probing | 61.4 | 90.2 | 74.8 | 95.5 | 90.2 | 46.9 | **55.8** | 73.5 | 81.5 | 90.1 | 82.1 | 69.4 | 80.8 | 39.1 | 35.9 | 40.1 | 65.0 | 20.3 | 26.0 | 14.3 | 27.6 | 33.5 | 58.2 | 0.05 |
| MLP-4 [24] | 54.9 | 87.4 | 71.4 | 99.5 | 89.1 | 39.7 | 52.5 | 70.6 | 80.5 | 90.9 | 76.8 | 74.4 | 80.7 | 60.9 | 38.8 | 40.2 | 66.5 | 9.4 | 21.1 | 14.5 | 28.8 | 31.2 | 57.7 | 4.04 |
| Partial [24] | 60.3 | 88.9 | 72.6 | 98.7 | 89.3 | 50.5 | 51.5 | 73.1 | 82.8 | 91.7 | 80.1 | 72.3 | 81.7 | 34.3 | 35.5 | 43.2 | 77.1 | 15.8 | 26.2 | 19.1 | 28.4 | 35.0 | 58.9 | 12.65 |
| Bias [23] | 73.1 | 86.8 | 65.7 | 97.7 | 87.5 | 56.4 | 52.3 | 74.2 | 80.4 | 91.6 | 76.1 | 72.5 | 80.1 | 47.3 | 48.5 | 34.7 | 66.3 | 57.6 | 36.2 | 17.2 | 31.6 | 42.4 | 62.1 | 0.25 |
| VPT-Shallow [24] | 78.0 | **91.3** | 77.2 | 99.4 | 90.4 | 68.4 | 54.3 | 79.9 | 80.1 | 93.9 | 83.0 | 72.7 | 82.5 | 40.8 | 43.9 | 34.1 | 63.2 | 28.4 | 44.5 | 21.5 | 26.3 | 37.8 | 62.9 | 0.05 |
| VPT-Deep [24] | **79.6** | 90.8 | **78.0** | **99.5** | 91.4 | 46.5 | 51.7 | 76.8 | 84.9 | 96.2 | 85.0 | 72.0 | 84.5 | 67.6 | 59.4 | 50.1 | 74.1 | 74.4 | 50.6 | 25.7 | 25.7 | 53.4 | 67.7 | 0.22 |
| ARC [3] | 62.5 | 90.0 | 71.9 | 99.2 | 87.8 | 90.7 | 51.1 | 79.0 | **89.1** | 95.8 | 84.5 | **77.0** | 86.6 | 75.4 | 57.4 | 53.4 | 83.1 | **91.7** | **55.2** | 31.6 | 31.8 | 59.9 | 72.6 | 0.27 |
| RLRR [29] | 66.1 | 90.6 | 75.5 | 99.3 | **92.1** | **90.9** | 54.7 | 81.3 | 87.1 | 95.9 | 87.1 | 76.5 | **86.7** | 66.0 | 57.8 | 55.3 | 84.1 | 91.1 | **55.2** | 28.6 | 34.0 | 59.0 | 73.0 | 0.41 |
| HTA | 72.0 | 89.6 | 76.4 | **99.5** | **92.1** | 87.8 | 55.5 | **81.8** | 86.7 | 96.3 | 87.5 | 76.3 | **86.7** | **85.0** | 62.2 | 53.7 | 84.3 | 89.1 | 52.4 | 27.6 | 36.4 | **61.3** | **74.2** | 0.23 |

Table 6: This table is extended from Table 5 in Section 4. LoRA and Adapter both follow the configurations from the A paper. In our implementation, the fully connected layers within the bottlenecks are replaced with Householder transformations. "($\cdot$)" indicates specific configuration information

| Methods | Natural | | | | | | | | Specialized | | | | | Structed | | | | | | | | | Mean Total | Params.(M) |
|---|---|---|---|---|---|---|---|---|---|---|---|---|---|---|---|---|---|---|---|---|---|---|---|---|
| | CIFAR-100 | Caltech101 | DTD | Flowers102 | Pets | SVNH | Sun397 | Mean | Camelyon | EuroSAT | Resisc45 | Retinopathy | Mean | Clevr-Count | Clevr-Dist | DMLab | KITTI-Dist | dSpr-Loc | dSpr-Ori | sNORB-Azim | sNORB-Ele | Mean | | |
| LORA($\mathbf{W}_q$, $\mathbf{W}_v$) | 67.1 | 91.4 | 69.4 | 98.8 | 90.4 | 85.3 | 54.0 | 79.5 | 84.9 | 95.3 | 84.4 | 73.6 | 84.6 | 82.9 | 69.2 | 49.8 | 78.5 | 75.7 | 47.1 | 31.0 | 44.0 | 59.8 | 72.3 | 0.29 |
| HTA($\mathbf{W}_q$, $\mathbf{W}_v$) | 71.2 | 93.3 | 72.5 | 99.3 | 91.2 | 82.7 | 56.6 | 81.0 | 85.3 | 94.9 | 82.5 | 75.7 | 84.6 | 80.8 | 62.9 | 50.2 | 78.9 | 77.4 | 51.1 | 29.7 | 45.4 | 59.6 | 72.7 | 0.09 |
| HTA($\mathbf{W}_q$, $\mathbf{W}_v$, $\mathbf{W}_{FC1}$, $\mathbf{W}_{FC2}$) | 71.7 | 93.1 | 70.9 | 99.2 | 90.5 | 86.6 | 55.8 | 81.1 | 87.6 | 94.9 | 82.5 | 76.2 | 86.3 | 80.8 | 60.1 | 51.0 | 82.0 | 86.9 | 52.2 | 32.9 | 46.0 | 61.48 | 73.9 | 0.28 |
| Adapter($\mathbf{W}_{FC1}$, $\mathbf{W}_{FC2}$) | 69.2 | 90.1 | 68.0 | 98.8 | 89.9 | 82.8 | 54.3 | 79.0 | 84.0 | 94.9 | 81.9 | 75.5 | 84.1 | 80.9 | 65.3 | 48.6 | 78.3 | 74.8 | 48.5 | 29.9 | 41.6 | 58.5 | 71.4 | 0.16 |
| HTA($\mathbf{W}_{FC1}$, $\mathbf{W}_{FC2}$) | 72.6 | 93.0 | 71.1 | 99.3 | 91.4 | 82.1 | 57.2 | 81.0 | 85.3 | 95.0 | 82.9 | 76.3 | 84.9 | 81.6 | 63.9 | 49.5 | 81.2 | 79.2 | 51.4 | 28.1 | 45.4 | 60.0 | 73.0 | 0.05 |

Table 7: This table is extended from Fig. 2 in Section 4.

| Methods | Natural | | | | | | | | Specialized | | | | | Structured | | | | | | | | | Mean Total | Params.(M) |
|---|---|---|---|---|---|---|---|---|---|---|---|---|---|---|---|---|---|---|---|---|---|---|---|---|
| | CIFAR-100 | Caltech101 | DTD | Flowers102 | Pets | SVNH | Sun397 | Mean | Camelyon | EuroSAT | Resisc45 | Retinopathy | Mean | Clevr-Count | Clevr-Dist | DMLab | KITTI-Dist | dSpr-Loc | dSpr-Ori | sNORB-Azim | sNORB-Ele | Mean | | |
| $D'$=0 | 73.0 | 90.1 | 71.8 | 99.3 | 91.1 | 83.4 | 53.7 | 80.3 | 82.3 | 94.2 | 82.7 | 73.7 | 83.2 | 77.3 | 61.6 | 49.2 | 80.0 | 81.7 | 53.3 | 28.1 | 41.2 | 59.1 | 72.0 | 0.15 |
| $D'$=1 | 76.6 | 94.3 | 72.5 | 99.3 | 91.3 | 86.2 | 56.5 | 82.4 | 87.6 | 95.7 | 85.0 | 75.7 | 86.0 | 82.6 | 63.3 | 52.5 | 81.0 | 84.5 | 52.6 | 34.5 | 47.3 | 62.3 | 74.7 | 0.22 |
| $D'$=2 | 75.5 | 94.2 | 73.0 | 99.3 | 91.2 | 85.7 | 56.3 | 82.2 | 88.0 | 95.2 | 84.6 | 76.1 | 86.0 | 82.2 | 63.1 | 52.1 | 80.2 | 85.5 | 52.3 | 33.9 | 47.5 | 62.1 | 74.5 | 0.30 |
| $D'$=4 | 74.6 | 93.6 | 72.1 | 99.3 | 91.4 | 88.3 | 56.1 | 82.2 | 88.0 | 96.3 | 85.6 | 76.4 | 86.6 | 81.8 | 64.9 | 53.6 | 82.3 | 84.8 | 53.6 | 34.4 | 47.2 | 62.8 | 75.0 | 0.44 |

enhanced mechanisms that can both identify and reduce such biases and unlawful information in the datasets.

