# OpenReview forum: "Efficient Adaptation of Pre-trained Vision Transformer via Householder Transformation"
_NeurIPS.cc/2024/Conference — NeurIPS 2024 poster_

### Official Review · Reviewer_PRgH · 2024-06-18

**Soundness:** 4
**Presentation:** 3
**Contribution:** 3
**Rating:** 6
**Confidence:** 3

**Summary:**

The study introduces a data-driven variant of LoRA-like Parameter-Efficient Fine-Tuning (PEFT) for Vision Transformers (ViTs), which learns the optimal bottleneck dimension (the rank of the fine-tuning weight matrix) instead of preconfiguring it for each layer. The authors propose learning three additional vectors: $v_{\text{left}}$, $v_{\text{right}}$, and a vector of diagonal parameters $D$ that define the rank. Inspired by Singular Value Decomposition (SVD), they employ Householder transformations $I-vv^\intercal$ on $v_{\text{left}}$, $v_{\text{right}}$ to construct orthogonal matrices. These learned vectors and parameters $v_{\text{left}}$, $v_{\text{right}}$, and $D$ can then be fused into the frozen weights in a LoRA-like manner. Jointly training $v_{\text{left}}$, $v_{\text{right}}$, and $D$ alongside an extremely low-rank LoRA branch (r=1) results in improved performance on downstream tasks.

**Strengths:**

The paper tackles the optimal rank determination problem for LoRA-based methods using a learning-based approach. Despite requiring an additional rank-1 LoRA branch to enhance accuracy, both experiments and the ablation study highlight the effectiveness of the proposed method, surpassing several state-of-the-art baselines.

**Weaknesses:**

While the proposed method is parameter-efficient, the discussion overlooks the actual training cost and latency. For instance, it would be beneficial to include theoretical or actual memory and computational costs.

**Questions:**

- What rank is applied to the LoRA-based baselines in Table 1? It would be beneficial to list these parameters in the appendix.

- What are the settings used in Table 5? Are the LoRA branches also applied in the HTA row?

- Could the authors include a section discussing theoretical and actual memory computation costs, such as FLOPS?

- Is it feasible to extend the method to Large Language Models?

**Limitations:**

As the author discussed in the limitation section.

---

> ### Author Rebuttal · Authors · 2024-08-06
>
> **Questions: What rank is applied to the LoRA-based baselines in Table 1? It would be beneficial to list these parameters in the appendix.**
>
> In Table 1, the rank for LoRA is set to 8. In subsequent versions, we will follow your suggestion and include the settings for the comparison methods in the appendix.
>
> **Questions: What are the settings used in Table 5? Are the LoRA branches also applied in the HTA row?**
>
> In Table 5, we followed the settings from FacT, where the bottleneck dimensions for both LoRA and Adapter were set to 8. Specifically, LoRA was applied only to the fully connected layers of Q and V in the MHA module (for a more direct comparison, we applied HTA in the Q and V fully connected layers as well). In our HTA module, we adhered to the previous experimental setup by setting the bottleneck dimension in the LoRA branch to 1. All other settings were kept consistent throughout the experiments.
>
> **Questions: Could the authors include a section discussing theoretical and actual memory computation costs, such as FLOPS?**
>
> Since our method involves applying Householder transformations to vectors and performing inverse SVD decomposition to multiply the left and right unitary matrices with singular values, it requires more computational steps and resources compared to LoRA. As a result, our method incurs slightly higher FLOPs and memory consumption.
>
> We conducted a simple calculation of FLOPs and memory usage using data of size (16, 3, 224, 224). The results are as follows: HTA (GFLOPs: 748; MEM: 5196M), LoRA (GFLOPs: 566; MEM: 4752M), Adapter (GFLOPs: 565; MEM: 4813M). The increase in FLOPs is primarily due to the Householder transformation, and the inverse SVD decomposition also introduces some additional computational overhead. Additionally, the increase in memory usage is mainly due to the extra gradient maps generated by these additional computational steps. In this experiment, the bottleneck dimensions for both Adapter and LoRA were set to 8. We will also follow your suggestion and include a separate section in future versions to discuss the theoretical and practical computational costs.
>
> **Questions: Is it feasible to extend the method to Large Language Models?**
>
> To validate the adaptability of HTA on large language models, we selected the Llama model from the NLP domain for further testing. We used the commonsense_15k dataset as the training set and the complete commonsense dataset as the test set. Comparing HTA with LoRA (rank=32) as the baseline, we found that despite HTA having only a quarter of the parameters of LoRA (rank=32), its performance was still comparable to that of LoRA (rank=32). This indicates that our method is also applicable to large language models.
>
> Thank you for your review and comments. Your recognition of our work, as well as the identified shortcomings and issues, is extremely helpful for both our current paper and future research. We will carefully consider your suggestions and incorporate them into our manuscript to enhance its overall quality.

---

> > ### Comment · Reviewer_PRgH · 2024-08-07
> >
> > Thank you for your responses. Could you kindly point out the tables that contain the experiments on common sense reasoning, object detection, and segmentation mentioned in the global response?

---

> ### Author Response · Authors · 2024-08-09
>
> Thank you for your prompt response. In our global response, we highlighted the major points addressing the questions but did not provide the detailed experimental results on common sense reasoning, object detection, and segmentation as mentioned. To clarify the additional experimental results requested, we have provided the tables below. Please note that the detailed experimental settings and illustrations can be found in the response to Reviewer #YBq9.
>
> | Method                             | Backbone          | Tuning Method | Params (M) | Acc  | Task                |
> | ---------------------------------- | ----------------- | ------------- | ---------- | ---- | ------------------- |
> | LlamaForCausalLM                   | llama             | LoRA (d=32)   | 25.16      | 69.8 | Commonsense Reasoning |
> |   LlamaForCausalLM     |    llama    | HTA  (d=6)    | 5.89       | 70.1 | Commonsense Reasoning |
>
> | Method                             | Backbone          | Tuning Method | Params (M) | mIoU | Task                |
> | ---------------------------------- | ----------------- | ------------- | ---------- | ---- | ------------------- |
> | Swin-Transformer+UperNet           | Swin-Transformer  | LoRA (d=8)     | 1.60       | 46.1 | Semantic Segmentation |
> |  Swin-Transformer+UperNet       |  Swin-Transformer    | HTA (d=1)           | 0.46       | 47.6 | Semantic Segmentation |
>
> | Method                             | Backbone          | Tuning Method | Params (M) | mAP  | Task            |
> | ---------------------------------- | ----------------- | ------------- | ---------- | ---- | --------------- |
> | Swin-Transformer+Cascade Mask RCNN | Swin-Transformer  | LoRA (d=8)    | 1.60       | 44.2 | Object Detection |
> |Swin-Transformer+Cascade Mask RCNN  | Swin-Transformer  | HTA (d=1)           | 0.46       | 47.2 | Object Detection |
> | ViTPose                            | ViT               | LoRA (d=8)    | 0.59       | 73.6 | Pose Estimation  |
> |ViTPose |     ViT           | HTA  (d=1)          | 0.19       | 74.1 | Pose Estimation  |

---

> > ### Comment · Reviewer_PRgH · 2024-08-09
> >
> > Thank you for the response. The additional experiments demonstrate the generality and effectiveness of the proposed method for language models as well as various tasks in computer vision. The paper addresses the issue of fixed bottleneck dimensionality in the LoRA-based method using Householder Transformation, which, to the best of my knowledge, is novel. While the method eliminates the need for bottleneck dimensionality tuning, my primary concern is that the experiments show comparable (or only marginally improved) performance compared to a fixed-rank LoRA (with a rank of 8) at the cost of increased computational resources. Although the proposed method uses significantly fewer parameters, the vectorized parameters are expanded into Householder matrices during training, which requires additional gradient computations. Based on the generality of the proposed method demonstrated in the additional experiments and the technical contribution of the paper, I will maintain my current rating.

---

> > > ### Author Response · Authors · 2024-08-09
> > > **Thank you for your recognition of our work**
> > >
> > > Thank you very much for your valuable comments and recognition of our work.

---

### Official Review · Reviewer_YBq9 · 2024-07-02

**Soundness:** 3
**Presentation:** 3
**Contribution:** 3
**Rating:** 5
**Confidence:** 2

**Summary:**

This paper proposes a new PEFT method by integrating Householder transformations, which enables the creation of adaptation matrices that have varying ranks across different layers, thereby offering more flexibility in adjusting pre-trained models.

**Strengths:**

S1: This paper proposes a new PEFT method based on Householder transformations.

S2: This method allows for layer-wise rank variations.

S3: The authors validated the effectiveness of the proposed method on classification tasks.

**Weaknesses:**

I am not an expert in the field of PEFT, it appears that the authors have proposed a new PEFT scheme. From an application perspective, we desire PEFT methods that can transfer the pre-trained models to downstream tasks with as few parameters as possible. In the experiments conducted by the authors, the proposed method achieved slight improvements with relatively fewer parameters. In fact, all methods utilize a minimal number of parameters.

In addition, the authors did not make comparisons in more complex tasks, such as object detection and semantic segmentation. Currently, numerous related studies employ a backbone based on the ViT [A,B,C]. Validation solely on classification tasks is somewhat limiting.


[A] Exploring Plain Vision Transformer Backbones for Object Detection.

[B] SegViT: Semantic Segmentation with Plain Vision Transformers.

[C] ViTPose: Simple Vision Transformer Baselines for Human Pose Estimation

**Questions:**

Is the proposed method effective in tasks such as object detection and semantic segmentation?

**Limitations:**

The novelty, contribution and experimental comparison of this paper all have certain shortcomings.

---

> ### Author Rebuttal · Authors · 2024-08-06
>
> We agree that all PEFT methods use a reduced number of parameters, and at this level, further reduction in parameter count has limited significance in practical applications. However, compared to previous low-rank-based PEFT methods such as LoRA and Adapter, our approach addresses a key issue: the fixed-rank limitation of the adaptation matrix due to preset bottleneck dimensionality. By using fewer parameters, we construct a rank-tunable matrix composed of three one-dimensional vectors, which can theoretically adjust its rank from 1 to full rank. This allows it to flexibly replace any low-rank matrix, thereby enhancing its adaptation capacity. This feature provides our method with great potential for application to more complex datasets.
>
> To further validate the effectiveness of our proposed method in object detection and semantic segmentation tasks, we conducted downstream experiments following the settings of Swin Transformer. For the object detection task, we used the COCO Object Detection (2017 val) dataset, selecting Swin-B as the backbone (with pre-trained weights on ImageNet1k) and using Cascade Mask RCNN as the decoder. The results show that our HTA method (mIoU=47.2) improved by approximately 3% compared to LoRA (mIoU=44.2).
> For the semantic segmentation task, we used the ADE20K Semantic Segmentation (val) dataset, again selecting Swin-B as the backbone (with pre-trained weights on ImageNet1k) and UperNet as the decoder. The results indicate that our HTA method (mIoU=47.6) outperformed LoRA (mIoU=46.1) by about 1.5%.
> Additionally, due to time constraints, we were unable to validate all the experiments you mentioned. Instead, we chose to validate using ViTPose, employing the ViTPose-B (simple) model and conducting experiments on the COCO Object Detection (2017 val) dataset. Using full fine-tuning with the MHSA module frozen as the baseline, we added LoRA and HTA to the MHSA module. The experimental results are as follows: baseline (mAP=72.8), LoRA (mAP=73.6), HTA (mAP=74.1). These results demonstrate the effectiveness of our method in downstream tasks and suggest that it could serve as an alternative to LoRA. It is important to note that all experiments, except for the ViTPose baseline, were independently reproduced by us. Apart from the HTA and LoRA modules, all other components were kept consistent to ensure the accuracy of the results.
>
> Thank you for your comments and questions. The issues you raised are very helpful for our future work. We will incorporate your suggestions to ensure that our work is useful across multiple fields.

---

> > ### Comment · Reviewer_YBq9 · 2024-08-10
> >
> > Thanks for the authors' responses, and I learned a lot through this paper. I have adjusted my score accordingly, and vote for acceptance of this paper.

---

### Official Review · Reviewer_HQtR · 2024-07-13

**Soundness:** 3
**Presentation:** 4
**Contribution:** 3
**Rating:** 7
**Confidence:** 3

**Summary:**

The paper presents a novel method, HTA, for efficiently adapting pre-trained transformers by applying the Householder transformation to a single vector to approximate the SVD for representing the adaptation matrix. By combining this with an additional rank-1 adaptation matrix, HTA achieves a reduction in the number of parameters while maintaining strong performance across various benchmarks, including FGVC and VTAB-1k, and outperforms existing methods in terms of efficiency and performance.

**Strengths:**

Originality:
This work incorporates the Householder transformations to construct orthogonal matrices, replacing the previously proposed low-rank adaptation matrix, and avoids the hyperparameter choice of the rank.

Quality:
The submission is technically sound with well-supported claims backed by extensive experimental results.

Clarity:
The paper is well-written and organized, making it easy to follow the methodology and results.

Significance:
1. This work addresses a challenging task and advances the state-of-the-art with high performance and reduced computational costs.
2. It may lead to further development of methodologies based on Householder transformations.

**Weaknesses:**

Originality:
Utilizing Householder transformations for matrix decomposition is not a brand-new idea.

Quality:
The improved performance of HTA over the state-of-the-art methods on several benchmarks is not very significant (<0.5%).

Clarity:
A scatter plot for accuracy versus the number of parameters tradeoff would give a clearer insight into HTA's performance.

Significance:
1. The challenge of solely relying on Householder transformations for adapting pre-trained transformers remains.
2. The proposed HTA removes the rank choice in previous adaptation methods but introduces a new rank choice in the additional adaptation matrix, although rank 1 might be sufficient for a good performance.

**Questions:**

1. Fig. 1c has a skip-connection, but Fig. 1d does not. Is it proposed to be like this?
2. The proposed HTA includes two feature paths, with both the Householder transformations path and the other down-up projection path being rank 1. It remains unclear why the additional down-up projection path would help significantly boost performance.
3. The results in Fig. 2 with dimension=0 are different from the ones in previous tables. What are the differences in settings between them?
For Swin, it would be better to include which specific Swin model is applied.
4. For Swin, it would be better to include which specific Swin model is applied.

**Limitations:**

The novelty and significance of HTA are somewhat limited. However, the extensive experimental results provide a robust foundation for the claims made in the paper. Overall, the work is well-executed and makes a valuable contribution to the field, justifying a recommendation for weak acceptance.

---

> ### Author Rebuttal · Authors · 2024-08-06
>
> **Weaknesses(Originality): Utilizing Householder transformations for matrix decomposition is not a brand-new idea.**
>
> The Householder transformation is indeed not a new concept in matrix decomposition; rather, it is a commonly used method. However, the core of our approach does not lie in the use of Householder transformations for matrix decomposition, but rather in transforming a one-dimensional Householder vector into a unitary matrix (analogous to the left and right unitary matrices in SVD decomposition) through the Householder transformation. In this way, we construct a rank-tunable matrix using three one-dimensional vectors (two Householder vectors and one singular value vector). This allows us to fine-tune models using rank-tunable matrices, freeing us from the constraints of fixed rank imposed by previous methods due to preset bottleneck dimensions.
>
> **Weaknesses(Quality): The improved performance of HTA over the state-of-the-art methods on several benchmarks is not very significant (<0.5%).**
>
> Our main contribution lies in providing new insights into the popular low-rank-based pre-trained model adaptation strategy, rather than merely improving fine-tuning performance. We aim to address the inherent limitations of low-rank-based adaptation strategies, such as LoRA and Adapter, which rely on a fixed rank of the adaptation matrix by pre-setting the bottleneck dimensionality. Instead, we construct a rank-tunable adaptation matrix by leveraging the idea of the Householder transformation. This method composes the adaptation matrix in a parameter-efficient manner using only three one-dimensional vectors. Our approach offers greater flexibility in constructing the adaptation matrix and provides a better trade-off between parameter count and adaptation capacity.
>
> **Weaknesses(Clarity): A scatter plot for accuracy versus the number of parameters tradeoff would give a clearer insight into HTA's performance.**
>
> Thank you for your suggestion. Following your advice, we have created a scatter plot to more clearly illustrate HTA's performance. The scatter plot is presented in Figure 2 in the rebuttal PDF.
>
> **Questions: Fig. 1c has a skip-connection, but Fig. 1d does not. Is it proposed to be like this?**
>
> In Figure 1d, we adhered to the original Adapter branch configuration, including the skip connection. However, for the sake of simplicity and aesthetics, we omitted the skip connection in the illustration. Following your suggestion, we have revised Figure 1d, and the updated figure is presented in Figure 2 of the rebuttal PDF.
>
> **Questions: The proposed HTA includes two feature paths, with both the Householder transformations path and the other down-up projection path being rank 1. It remains unclear why the additional down-up projection path would help significantly boost performance.**
>
> In the Householder transformation branch, the left and right unitary matrices are derived from Householder mappings, resulting in orthogonally symmetric unitary matrices. Although the Householder matrix itself possesses mapping capabilities and can indirectly represent rotational characteristics as the Householder vector is continuously optimized during training, its inherent symmetry limits its ability to perfectly replicate arbitrary unitary matrices, imposing certain constraints. To address this, we introduced an additional matrix to break these constraints. Considering the parameter count, we opted for a down-up projection with a dimension of 1 to resolve this issue.
>
> **Questions: The results in Fig. 2 with dimension=0 are different from the ones in previous tables. What are the differences in settings between them?**
>
> We need to clarify that in the previous tables, we used dimension=1 as our setting, and it can be observed that the results for dimension=1 in Figure 2 are consistent with those in the previous tables. In the ablation study presented in Figure 2, setting dimension to 0 is intended to validate the performance of the Householder transformation branch itself and the contribution of the down-up projection branch. In future versions, we will provide a more precise description of the settings for each component.
>
> **Questions: For Swin, it would be better to include which specific Swin model is applied.**
>
> Specifically, we used Swin-B as our fine-tuning model, with the pre-trained weights obtained from the official swin_base_patch4_window7_224_22k.pth, which was trained on ImageNet21k. The details are as follows:
>     "base" refers to the base model, which strikes a balance between performance and computational complexity.
>     "Patch4" indicates that the input image is divided into 4x4 patches for processing.
>     "window7" denotes the use of a 7x7 window size for the window attention mechanism during each stage of computation.
>     The input size of 224 refers to the image size of 224x224 pixels used during pre-training.
>
> Thank you very much for your thorough review and valuable comments. Your feedback has been instrumental in helping us refine our work, and we greatly appreciate the time and effort you have dedicated to this process. We look forward to further improving our research based on your insights.

---

> ### Comment · Reviewer_HQtR · 2024-08-13
>
> Thank you for the detailed response. I have updated my vote to accept.

---

### Official Review · Reviewer_6gjh · 2024-07-13

**Soundness:** 4
**Presentation:** 4
**Contribution:** 4
**Rating:** 7
**Confidence:** 4

**Summary:**

The paper presents a new parameter efficient fine tuning (PEFT) technique for vision transformer (ViT) models. The work utilizes the intuition of creating an adaptation matrix for fine-tuning from a popular dimensionality reduction technique named singular value decomposition (SVD) which is named Householder transformation based Adapter (HTA).

HTA can be utilized to create adaptation matrices to accommodate layer wise variations. Extensive empirical evaluations have been performed over various popular vision benchmarks. Code has been made available for reproducibility.

**Strengths:**

HTA presented in this work is a new PEFT technique to avoid full fine-tuning for large vision models. The empirical evaluations are extensive and rigorous and HTA is benchmarked against widely popular PEFT techniques. HTA achieves great performance similar to established previous state-of-the-art thresholds with gains in terms of trainable parameter efficiency.

The readability of the work is great with great visualizations and empirical evaluations have been crisply presented. The writing structure and language are easily digestible for the reader.

**Weaknesses:**

No weaknesses apart from the limitations mentioned in Section 5 by the authors

**Questions:**

Do you think the HTA idea is adaptable to other modalities for PEFT?

---

> ### Author Rebuttal · Authors · 2024-08-06
>
> **Questions: Do you think the HTA idea is adaptable to other modalities for PEFT?**
>
> To validate the adaptability of HTA in other modalities, we selected the large model Llama from the NLP domain for further testing. Due to time constraints, we only used the smaller commonsense_15k dataset as the training set and the complete commonsense dataset as the test set. We used LoRA (rank=32) as a baseline to compare its performance with that of HTA. Despite HTA having only a quarter of the parameters of LoRA (rank=32), its performance was still comparable to that of LoRA (rank=32). This indicates that our method is not only applicable to image modalities but also to other modalities such as text.
>
> We greatly appreciate your suggestion to explore this research direction and will continue to investigate its potential in other modalities in future studies. Thank you very much for your valuable feedback.

---

> > ### Comment · Reviewer_6gjh · 2024-08-12
> >
> > thank you for your response.

---

### Author Rebuttal · Authors · 2024-08-07

We sincerely appreciate the reviewers and ACs for their thorough and valuable comments on our manuscript and their recognition of our work. We aim to address the shared concerns raised by the reviewers and provide a unified response below. Additionally, we have offered detailed replies to the specific questions posed by each reviewer within separate rebuttal windows.

**Limited performance improvement and whether further reducing the parameter count is practical given that other PEFT methods already have sufficiently small parameter sizes.** **Re:** Our primary contribution lies not merely in improving fine-tuning performance or reducing parameter count, but in offering new insights into popular low-rank-based pre-trained model adaptation strategies. Our goal is to address the inherent limitations of low-rank adaptation strategies, such as LoRA and Adapter, which rely on fixed-rank adaptation matrices with predefined bottleneck dimensions. To overcome this, we have utilized the concept of Householder transformations to develop an adaptable matrix with adjustable rank. This approach requires only three one-dimensional vectors to efficiently construct the adaptation matrix, providing greater flexibility in matrix design and achieving a better balance between parameter size and adaptability.

**Whether HTA is effective in downstream tasks, other modalities, and large language models.** **Re:** We conducted validations across different domains, including common-sense reasoning, object detection, and semantic segmentation, using models like Llama, Swin Transformer, and ViTPose. The experimental results demonstrate that our method often achieves comparable or even superior performance with less data compared to LoRA, effectively positioning it as a superior alternative. This further validates the effectiveness and generalizability of our approach.

---

### Decision · Program_Chairs · 2024-09-25

**Decision:**

Accept (poster)

**Comment:**

The reviewers unanimously voted for acceptance. They appreciated the technical originality, clarity and extensive experiments that support the efficacy of the proposed approach. After carefully reading the reviews and the rebuttal, this meta-reviewer concurs with the reviewers' recommendations.